



# Mitigation of Model Bias Influences on Wave Data Assimilation with Multiple Assimilation Systems Using WaveWatch III v5.16 and SWAN v41.20

Jiangyu Li[1,4] and Shaoqing Zhang[*1,2,3,4]

[1]Key Laboratory of Physical Oceanography, MOE, China; Ocean University of China, Qingdao, 266100, China
[2]Pilot National Laboratory for Marine Science and Technology (QNLM), Qingdao, 266100, China
[3]International Laboratory for High-Resolution Earth System Model and Prediction (iHESP), Qingdao, 266100, China
[4]The College of Oceanic and Atmospheric Sciences, Ocean University of China, Qingdao, 266100, China

*Correspondence to*: Shaoqing Zhang (szhang@ouc.edu.cn)

**Abstract.** High-quality wave prediction with a numerical wave model is of societal value. To initialize the wave model, wave data assimilation (WDA) is necessary to combine the model and

observations. Due to imperfect numerical schemes and approximated physical processes, a wave model is always biased in relation to the real world. In this study, two assimilation systems are first developed using two nearly independent wave models; then, "perfect" and "biased" assimilation frameworks based on the two assimilation systems are designed to reveal the uncertainties of WDA. A series of "biased" assimilation experiments is conducted to systematically examine the adverse

impact of model bias on WDA. A statistical approach based on the results from multiple assimilation systems is explored to carry out bias correction, by which the final wave analysis is significantly improved with the merits of individual assimilation systems. The framework with multiple assimilation systems provides an effective platform to improve wave analyses and predictions and help identify model deficits, thereby improving the model.

## 1 Introduction

Ocean waves, referring to the ocean surface gravity waves driven by wind, are important physical processes in the study of multiscale coupled systems. Many studies show that ocean waves are necessary for upper ocean mixing processes, whether in small-scale coastal simulations or large-scale





global climate simulations (e.g., Babanin et al., 2009; Huang and Qiao, 2010; Qiao et al., 2004, 2010).

The existence of ocean waves can modify the structures of both atmospheric and marine boundary layers by providing sea surface roughness, wave-induced bottom stress, breaking wave-induced mixing, and so on, which ultimately influence air-sea momentum and heat exchange. Therefore, ocean waves are an important component in atmosphere-ocean interaction flux processes (e.g., Chen et al., 2007; Doyle, 2002; Liu et al., 2011; Warner et al., 2010). In addition, the study of ocean waves can reduce

and prevent marine disasters and provide guidance for development of the social economy (e.g., Folley and Whittaker, 2009; Rusu, 2015; Wei et al., 2017). Thus, studying ocean waves is of great scientific and social significance.

At present, ocean wave observational techniques are constantly being improved (e.g., Daniel et al., 2011; Hisaki, 2005). Except for traditional buoy observations (e.g., Mitsuyasu et al., 1980; Rapizo et al.,

2015; Walsh et al., 1989), satellites can provide much near real-time wave observational information, which is beneficial for understanding the state of ocean waves (e.g., Gommenginger et al., 2003; Lzaguirre et al., 2011; Queffeulou P, 2004). However, observations always represent scattered samples in time and space in the real world and therefore do not represent the complete three-dimensional structure and temporal evolution of real world waves.

Numerical wave models are a powerful tool for studying the physical processes of ocean waves and predicting future wave states. Following the development of the previous two generations, third-generation wave models, such as WAve Modeling (WAM) (WAMDI Group, 1988), WaveWatch III (WW3) (Tolman, 1991), Simulating Waves Nearshore (SWAN) (Booij et al., 1999), and MArine Science and Numerical Modeling (MASNUM) (Yang et al., 2005), integrate the spectral action balance

equation describing the two-dimensional ocean wave spectrum evolution without additional *ad hoc* assumptions regarding the spectral shape, and these third-generation models are more robust for arbitrary wind fields than previous models. However, there are generally three error sources in wave models. One error source is from an incomplete understanding of the physical processes, approximate expressions of the numerical discretization schemes and so on, which causes systematic errors that are

usually referred to as wave model bias. The second error source is due to inaccurate wind forcings of wave models. The third error source is from the initial condition uncertainties, which can grow due to nonlinearity of the model equations during model forwarding. In this sense, the model simulated waves do not represent the real world either.





Given the scattering nature of observational information and the approximate characteristics of wave
modeling, wave model data assimilation (WDA) is necessary to combine the advantages of both the
model and observations. WDA optimizes the model initial conditions to produce more accurate wave
forecasts and produces more accurate evolution of the 3-dimensional wave states to elucidate the
underlying mechanisms; this approach dates back to the 1980s (e.g., Esteva, 1988; Janssen et al., 1989).
Since then, many advanced WDA methods have been developed (e.g., Abdalla et al., 2013; Bauer et al.,
1996; Greenslade and Young, 2004; Jesus and Cavaleri, 2015; Lionello et al., 1992; Sun et al., 2017;
Vorrips et al., 1999), and their applications have been assessed (e.g., Francis and Stratton, 1990; Heras
et al., 1994; Stopa and Cheung, 2014). Furthermore, various observation types, such as buoy, radar and
satellite, have been applied to WDA (e.g., Bhatt et al., 2005; Breivik et al., 1998; Feng et al., 2006;
Greenslade, 2001; Hasselmann et al., 1997; Qi and Cao, 2016; Voorrips, 1999; Waters et al., 2013),
and the wave forecasts have also been directly addressed (e.g., Almeida et al., 2016; Emmanouil et al.,
2012; Lionello et al., 1995; Qi and Fan, 2013; Sannasiraj et al., 2006; Voorrips, 1999; Wang and Yu,
2009; Zhang et al., 2003).

Due to the approximate nature of the numerical discretization and physical processes, a systematic
difference between a model and the real world (i.e., model bias) exists. As noted by Zhang et al. (2012),
since model bias is not well defined in observational space, the influence of model bias on data
assimilation is a challenging research topic. Alternatively, one can simulate model bias using a pair of
models and study the adverse impacts on data assimilation. Inspired by previous work (e.g., Dee, 2005;
Zhang et al., 2012), here, we use a simple data assimilation scheme with two wave models (WW3 and
SWAN) to explore the influences of different error sources on WDA. The adverse impacts of wind
forcing errors and initial condition uncertainties as well as wave model bias on WDA are studied first,
and then two simple statistical methods for bias correction are developed to mitigate assimilation errors
and improve wave analysis.

This paper is organized as follows. After the introduction, the methodology is presented in section 2,
including a brief description of the employed models and observations, development of the two WDA
systems using the WW3 and SWAN models, as well as the design of experiments throughout the study.
Section 3 presents the model bias analysis and the adverse impacts of model bias on WDA. In section 4,
the method used to mitigate model bias influences on wave assimilation is explored. Finally, the
discussion and conclusion are given in section 5.





## 2 Methodology

### 2.1 Models and data

### 2.1.1 Three models

In the wave models, the variance spectrum or energy density $E(\sigma, \theta)$ is a quantity that represents the wave energy distribution in the radian frequency ($\sigma$) and propagation direction($\theta$). Without ambient ocean currents, the variance or energy of a wave package is conserved. However, if the current is

involved, due to the work done by the current on the mean momentum transfer of waves (Longuet-Higgins and Stewart, 1961, 1962), the energy of a spectral component is no longer conserved. In general, an action density spectrum defined as N=E/$\sigma$ is considered within the models. Then, the governing equation of the wave model can be written as follows:

$$\frac{\partial N}{\partial t} + \nabla_{\vec{x}} \cdot (\overrightarrow{c_g} N) + \frac{\partial c_\sigma N}{\partial \sigma} + \frac{\partial c_\theta N}{\partial \theta} = \frac{S_{tot}}{\sigma}. \tag{1}$$

The left-hand side is the kinematic part of this equation. The second term describes the wave energy propagation in two-dimensional geographical space denoted by $\overrightarrow{x}$. $\overrightarrow{c_g}$ is the group velocity that follows the dispersion relation. The third term represents the effect of shifting the radian frequency due to variation in depth. The fourth term represents the depth-induced refraction. $c_\sigma$ and $c_\theta$ are wave velocities in the frequency $\sigma$ and direction $\theta$, respectively. On the right-hand side, $S_{tot}$ is the

nonconservative source/sink term representing all physical processes that generate, dissipate, or redistribute wave energy. Typically, there are three important physical processes that contribute to $S_{tot}$, which include the atmosphere-wave interaction, nonlinear wave-wave interaction, and wave-ocean interaction. In a shallow-water case, additional processes must be considered, such as wave-bottom interaction, depth-induced breaking, and triad wave-wave interaction.

In this study, we use three advanced third-generation spectrum models. The first one is WAve Modeling (WAM, Cycle 4.5.4) developed from the Sea Wave Modelling Project (SWAMP). For the first time, WAM creatively overcomes the shortcomings of the first- and second-generation wave models, such as the numerical problems and the restrictions on the spectral shape, and is available for all wind fields and extreme situations.

The second model is WaveWatch III (WW3, version 5.16), which is provided online by the National Center for Environmental Prediction (NCEP). In terms of the major aspects such as governing equations, program structures, as well as numerical and physical approaches, WW3 is different from its





predecessors (WW2 developed at the NASA Goddard Space Flight Center, and WW1 developed at Delft University of Technology, Netherlands), as WW3 has a more reasonable wind-wave physical

mechanism.

The third model is Simulating WAves Nearshore (SWAN, version 41.20) provided by Delft University of Technology, Netherlands. SWAN model employs a fully implicit finite differencing scheme, which is unconditionally stable and is more focused on wave propagation processes in shallow water. The model has been successfully applied to simulations of waves in coastal areas, lakes and estuaries, and

so on.

### 2.1.2 Model configurations

Three wave models use two-dimensional spectral space containing 29 frequencies that cover from .035 Hz to .555 Hz with a logarithmic distribution and 24 equidistant directions. The geographic space is from 180° W to 180° E in the zonal direction and 75° S to 75° N in the meridional direction with a

1°×1° grid resolution. The topography in this study is taken from the high-resolution ETOPO1 dataset provided by NOAA (website: https://www.ngdc.noaa.gov/mgg/global/). The wind forcing has two sources, both of which have 6-hour time intervals. The first dataset is the ERA-Interim reanalysis from European Centre for Medium-Range Weather Forecasts (ECMWF), with a resolution of .125°×.125° (http://apps.ecmwf.int/datasets/data/interim-full-daily/). The second dataset is the CFSRv2 dataset from

NCEP, with a resolution of .205° (longitude) ×.204° (latitude) (https://rda.ucar.edu/datasets/ds094.1/). The time step of all three models is 15 minutes. All relevant parameters are set to be identical for every wave model (WW3, SWAN, and WAM) in this study.

### 2.1.3 Data

The AVISO (Archiving, Validation and Interpolation of Satellite Oceanographic) data

(https://www.aviso.altimetry.fr/en/data/products/) are the satellite observational products used in this study. For ocean waves, the AVISO has two satellite altimetry products: along-track data and gridded data. The along-track data are used as the observational data source for the wave data in the simulation (sampled from "truth" in the "twin" experiments). The gridded data are used to validate the wave simulation and assimilation (1°×1° resolution with 1-day time intervals). During wave simulation, the

significant wave height (SWH) is used as a basic observational variable for data assimilation, which is provided from three ongoing satellites: Jason-2, Jason-3, and Satellite for Argos and ALtiKa (SARAL).





**Figure 1** shows one-cycle ground orbit by taking Jason-2 and SARAL as examples. Jason-3 is the successor of Jason-2, and both satellites share the same orbit.

**2.2 Different modeling strategies in WW3 and SWAN**

Since the observations are only a sample of real world information, the model bias (i.e., systematic difference between a numerical wave model and the real world) is not well defined against the real world. In this study, we use the systematic difference between the WW3 and SWAN models to simulate the model bias and study the influences on wave data assimilation (WDA).

First, let us distinguish the difference in physical and numerical aspects to comprehend the causes of
"bias" between these two models. In general, WW3 addresses global scales, and SWAN is more applicable in shallow water. Although the two models have most of the same physical processes, such as the wind input and nonlinear wave-wave interactions, each uses a different parameterization scheme. For example, the nonlinear wave-wave interactions in SWAN include the Discrete Interaction Approximation (DIA) (Hasselmann et al., 1985) and the Webb-Resio-Tracy (Resio and Perrie,
1991;Van Vledder, 2006; Webb, 1978), while there are more choices in WW3, such as the Generalized Multiple DIA (Toman, 2004, 2013), the Two-Scale Approximation and Full Boltzmann Integral (Perrie et al., 2013; Perrie and Resio, 2009; Resio et al., 2011; Resio and Perrie, 2008), as well as the Nonlinear Filter scheme (Tolman, 2011). In numerical aspects, there exist different implementation strategies such as the differencing method, which also contributes to bias.

**2.3 Two data assimilation systems using WW3 and SWAN**

To explore the model bias influences on WDA, we develop two data assimilation systems based on WW3 and SWAN in this study.

Generally, based on the program structure of wave models, we insert the assimilation module between calculations of the two-dimensional wave spectrum and outputs of wave parameters so that at the
assimilation time, we call on the assimilation module to update the spectrum and SWH. When building the data assimilation systems, we need to consider the different structures of parallelism method, data storage, and information exchange in WW3 and SWAN models as noted in section 2.2.

To clearly demonstrate the influences of model bias on WDA and minimize its adverse impact, the analysis scheme in both assimilation systems is optimal interpolation (OI), which also is low cost and
easy to operate. We implement the OI analysis in three to four steps. The first step uses two Gaussian





convolutions of the background and observed SWHs to compute the observational increment of SWH at the observational location. The second step projects the SWH observational increment onto the model grids centered at the observational location but within an impact radius using linear regression. The third step transforms the analyzed SWH to update the spectrum of model waves. The fourth step

corrects wind forcing using the observational SWH.

Step 1: Computing observational increment by convolution of two Gaussians

Starting from the idea of the ensemble adjustment Kalman filter (Andersen, 2001), an observational increment at the observational location $k$, $\Delta H_k^O$ ($H$ represents SWH), is computed by the convolution of two Gaussians of the model background and observation, which can usually be obtained from model

ensemble members and observational samples. $\Delta H_k^O$ is formulated as follows (Zhang et al, 2007):

$$\Delta H_k^O = \frac{\frac{1}{(\sigma^M)^2}\bar{H}^M + \frac{1}{(\sigma^O)^2}\bar{H}^O}{\frac{1}{(\sigma^M)^2} + \frac{1}{(\sigma^O)^2}} + \frac{\Delta H_k^M}{\sqrt{1+(\frac{\sigma^M}{\sigma^O})^2}} - H_k^M. \tag{2}$$

Here, the first and second terms on the right-hand side adjust the ensemble mean and ensemble spread, respectively, and $\Delta H_k^M$ represents the prior model spread. Superscripts $O$ and $M$ denote the observation and prior quantity estimated by the model, respectively. $\sigma$ is the corresponding error standard deviation.

The overbar denotes the ensemble mean. In this simplified case, we specify $\sigma^M = 0.6\,m$, $\sigma^O = 0.25\,m$, as in a previous study (Qi and Fan, 2013), and use single model and observational values as the ensemble mean.

Step 2: Regressing the observational increment onto model grids

The second step projects the observational increment $\Delta H_k^O$ onto the related model grids using

background error covariance, which is a key step in the analysis. To simplify the problem and improve the computational efficiency, many studies use a flow-independent distance function to sample the background error covariance for computing the analysis increment at the model grid $i$, $\Delta H_i^A$, as $\Delta H_i^A = (\sigma_i^s)^2 \exp\left(-(\frac{d_{i,k}}{L})\right) \times \Delta H_k^O$. Usually, such an expression is only a symmetrical approximation of the correlation function and cannot represent the spatial structure and propagation characteristics of waves.

Here, we modify the covariance formula to increase its representation for wave structure by superimposing a statistical correlation coefficient into the formula. After analysis, the equation becomes

$$\Delta H_i^A = \frac{\sigma_i^s}{\sigma_k^s} r_{i,k}^s \exp\left(-\left(\frac{d_{i,k}}{L}\right)\right) \times \Delta H_k^O, \tag{3}$$





where $L$ is the characteristic length and $d_{i,k}$ is the distance between the model grid $i$ and observational

point $k$. When $d_{i,k}$ is larger than the impact radius $R$, there is no observational impact on this model

point from observation $k$. All variables with superscript $s$ represent the model statistics from free model

control results. For example, $r_{i,k}^s$ is the SWH covariance between the model grid $i$ and observation $k$,

which is evaluated from the model data time series in corresponding experiments. To ensure the local

characteristics of ocean waves, in this study, the characteristics length $L$ and impact radius $R$ (or the

largest $d_{i,k}$) are the same, causing this incremental projection to reach to the e-folding scale. Referring

to previous studies (e.g., Lionello and Gunther, 1992; Qi and Fan, 2013), we tested different values of

$L$ and $R$ as 300 km, 800 km, and 1000 km and found no essential improvement with larger $L$ and $R$

values. Trading-off with computational efficiency, we set $L$ and $R$ as 300 km throughout this study. As

shown in **Fig. 2**, the new covariance represents more wave physics, i.e., the correlation has more

asymmetrical and wave-dependent characteristics.

Step 3: Transforming the SWH to wave spectrum

The assimilation SWH $H_i^A$ is a sum of the prior $H_i^M$ and the analysis increment from step 2 ($H_i^A = H_i^M + \Delta H_i^A$). In the wave model, the form of ocean waves is a two-dimensional wave spectrum that is

distributed over frequency and phase. Thus, transforming the assimilation SWH to wave spectrum is

necessary to update other wave parameters. Following the previous study (Qi and Fan, 2013), we

assume that the change in wave spectrum is proportional to the energy change that is expressed by the

square of SWH. Then, the analyzed spectrum $S_i^A(f, \theta)$ can be written as follows:

$$S_i^A(f, \theta) = \left(\frac{H_i^A}{H_i^M}\right)^2 S_i^M(f, \theta), \tag{4}$$

where $f$ is the wave frequency and $\theta$ is the phase direction.

Step 4: Correcting wind forcing using SWH data

If the assimilation only adjusts the wave spectrum as described in Step 3, the updated spectral structure

may be quickly overwritten by erroneous wind. In this step, we describe a simple scheme using the

observed SWH data to correct the wind forcing. Starting from a first guess of wind (the ERA-Interim

reanalysis, for instance), the analyzed wind $W_{i,j}^A$ at model grid $(i, j)$ can be written as follows:

$$W_{i,j}^A = W_{i,j}^M + \Delta W_{i,j}, \tag{5}$$

where $W$ represents either the $u$ or $v$ component of wind. $\Delta W_{i,j}$ is the corrected wind increment

transformed from the updated SWH. While the details of the transformation scheme can be found in



Lionello et al. (1992, 1995), we comment on certain aspects relevant to our study. Regardless of boundaries, in general, the energy of ocean waves is determined by the wind speed and duration, which

can also be expressed by SWH. In that sense, a function equation can be built, in which the left-hand side is an expression of wind speed and duration, while the right-hand side is an expression of SWH, and they are balanced through wave energy. Then, the analyzed wind speed can be resolved under the assumption that the duration is same in the both prior and analyzed fields.

With respect to the configuration of wave model data assimilation, the model time step is 15 minutes

and the assimilation interval is 1 hour. At the assimilation time, we assimilate the along-track observations within a 1-hour time window centered at the time. After 10 days, all the observations will cover the global area. The wind data from the reanalysis products (ERA-Interim and NCEP-CFSR in this case) are available every 6 hours. To incorporate the wind correction into the wind forcing of the model, we distribute the wind correction to the adjacent two time-levels of wind data. As the process is

looped forward as the wave model state is updated, the wind forcing is adjusted through the SWH assimilation.

## 2.4 Experimental design

Throughout this study, we use the symbol $MA_{O(s)}{}^{WF}$ as the name for the assimilation experiment. Here, "*MA*" stands for the "assimilation model" and the subscript "*O(s)*" (resp. superscript "*WF*") represents

the observing system (resp. wind forcing) in the assimilation. The wind forcing is either the ECMWF ERA-Interim (hereafter known as ERAI) or NCEP-CFSR wind (hereafter known as CFSR). Wind forcing can also be corrected by observations of SWH (under this circumstance, the superscript "WF" is replaced by "ASSW"). The observations used in the assimilation could be the model data but are projected on the along-track points of satellite(s) if being used for the twin experiments. Under this

circumstance, "O" represents "model that produces observations" and "(s)" represents the used satellite tracks (J2-Jason-2, J3-Jason-3, and SA-SARAL, for instance). Otherwise, in the real-data assimilation experiments, the subscript "*O(s)*" directly lists the satellites that measure the SWH.

### 2.4.1 Twin experiments

Twin experiments refer to a type of Observing System Simulation Experiment (OSSE), in which a

model simulation is used to define the "true" solution of a data assimilation problem, and the other model simulation is used to start the assimilation. The "observations" are samples of the "truth" with





some white noise to simulate the observational errors. When the "truth" and assimilation are conducted by different (resp. identical) models, the framework is a "biased" (resp. "perfect") model twin experiment. Within a twin experiment framework, any aspect of assimilation skills can be measured as

the degree to which the "truth" is recovered through the assimilation.

a) Perfect twin experiment

In a perfect twin experiment, we assume that the assimilation model and the observation are unbiased, i.e., both the instrument measuring and numerical modeling processes are sampling the same stochastic dynamical system. Such sampling only has random sampling errors without any systematic difference

(bias). We can build this perfect model framework by using the same model to produce the "truth" as the assimilation model but with different initial conditions and wind forcings.

The "observations" from the observational time window (1 hour) centered at the assimilating time can be created by sampling the "truth" SWH with the tracks of the Jason-2, Jason-3 and SARAL satellites, which will cover the global area in 10 days. In this circumstance, if WW3 (resp. SWAN) is used as the

assimilation model, the "truth" is produced by the same WW3 (resp. SWAN) model. In the assimilation, we may start the model with different initial conditions and/or wind forcings to examine the influences of initial errors and wind forcing errors on the wave assimilation. Such a perfect twin experiment can be named $WW3_{WW3(s)}{}^{WF}$ or $SWAN_{SWAN(s)}{}^{WF}$.

b) Biased twin experiment

To study the impact of model errors on wave assimilation, we use two models to design a "biased" twin experiment. Again, due to the scattering nature of the observations, it is difficult to obtain a complete picture of the model bias against the real world. Given the difference between the WW3 and SWAN models described in section 2.2, we use these two models and their assimilation systems here to simulate the model bias and examine its influences on the WDA. We use the ERA-Interim reanalysis

wind to force the WW3 (resp. SWAN) to produce the "truth" and "observations" but use the SWAN (resp. WW3) assimilation system to assimilate the "observations." The degree to which the "truth" produced by different model-based assimilation systems is recovered by assimilating the "observations" is an assessment of the model bias influences on the WDA. Such a "biased" twin experiment can be named $WW3_{SWAN(s)}{}^{WF}$ or $SWAN_{WW3(s)}{}^{WF}$.

Under the biased twin experiment framework, we also conduct experiments to examine the impacts of observing systems on wave assimilations by increasing the observational information based on multiple satellite tracks. For example, we can examine the assimilation results of $WW3_{SWAN(J2)}{}^{WF}$,





$WW3_{SWAN(J2+J3)}{}^{WF}$, and $WW3_{SWAN(J2+J3+SA)}{}^{WF}$ (resp. $SWAN_{WW3(J2)}{}^{WF}$, $SWAN_{WW3(J2+J3)}{}^{WF}$, and $SWAN_{WW3(J2+J3+SA)}{}^{WF}$) to understand the impacts of observing systems on different model-based

assimilations.

### 2.4.2 Real-data assimilation experiments

In this study, we also conduct real-data assimilation experiments using WW3 and SWAN assimilation systems with real track data from the Jason-2, Jason-3 and SARAL satellites. Through real-data assimilation experiments with different model-based assimilation systems, we can 1) increase our

understanding of the influences of model errors on the WDA and 2) study the method to reduce the model error influences on the assimilation results. The real-data assimilation experiments can be directly named, e.g., $WW3_{J2+J3+SA}{}^{WF}$ or $SWAN_{J2+J3+SA}{}^{WF}$.

### 3 Error sources in wave models and WDA

### 3.1 Influences of initial and wind forcing errors

Usually, wave numerical simulation can be improved by three methods: 1) reducing the errors in the initial conditions, 2) enhancing the accuracy of the wind forcing, and 3) improving the representation of the wave model and its parameterization.

In this section, we use perfect model twin experiments (as described in section 2.4.1) to exclude model errors and explore the impact of wind forcings and initial conditions on the wave simulations. To

compare the performances of the WW3 and SWAN models, we conduct separate experiments with these two models. The "truth" and model control runs are two basic experiments of the perfect twin experiment framework. We use the ERA-Interim wind to drive WW3 (resp. SWAN) and generate a long time series of model states as the "truth," which is called $WW3^{ERAI}$ (resp. $SWAN^{ERAI}$) for the WW3 (resp. SWAN) perfect model twin experiment. The "observations" are created by interpolating the

corresponding "truth" SWH onto the along-track points of satellite orbits. Then, we use the NCEP-CFSR wind to force WW3 (resp. SWAN), called the model control $WW3^{CFSR}$ (resp. $SWAN^{CFSR}$), and the data assimilation is named $WW3_{WW3(s)}{}^{CFSR}$ (resp. $SWAN_{SWAN(s)}{}^{CFSR}$). Starting from an independent initial condition produced by the model control, we can conduct the assimilation with the ERA-Interim or NCEP-CFSR wind forcing. The error verification of the assimilation results against the "truth"





simulation compared to the error of the model control is an evaluation of the initial error or/and wind forcing error influences on the WDA. All perfect model twin experiments are listed in **Table 1.**

First, we conduct two sets of model control experiments $WW3^{CFSR}$ and $SWAN^{CFSR}$ for 80 days (from December 2017 to February 2018). To explore the effect of the initial conditions, we perform the model spin-up for a long time to adequately reach a steady state. Then using the 45$^{th}$-day model states

as the initial conditions, we conduct one more model simulation and data assimilation experiments for each model system as $WW3^{ERAI}$ and $WW3_{WW3(J2)}{}^{ERAI}$ as well as $SWAN^{ERAI}$ and $SWAN_{SWAN(J2)}{}^{ERAI}$. The root mean square errors (RMSEs) of these experiments against the "truth" are shown in **Fig. 3** as the red (for $WW3^{CFSR}+WW3^{ERAI}$ and $SWAN^{CFSR}+SWAN^{ERAI}$) and pink (for $WW3^{CFSR}+WW3_{WW3(J2)}{}^{ERAI}$ and $SWAN^{CFSR}+SWAN_{SWAN(J2)}{}^{ERAI}$) lines.

The SWH RMSE is approximately .34 meters in the WW3 or SWAN model control with the NCEP-CFSR wind. Once the wind forcing is changed to the "perfect" wind (the ERA-Interim) on the 45$^{th}$ day, the RMSE quickly drops and is close to zero after approximately 10 days, and the SWAN model takes longer to accomplish this change than the WW3. If data assimilation is added, the RMSE reduces much faster than the model controls (roughly half of the time scale of the correct wind forcing). From the

analyses above, we learned, 1) in wave models, the wind forcing plays an important role and an incorrect wind forcing could be a significant error source of WDA; and 2) the WDA can rapidly reduce the initial error and improve the predictability of a wave model even when it is forced by an accurate wind forcing.

### 3.2 Impact of the observational system

In this section, using the same model states (at the 45$^{th}$ day) in the corresponding model control as in section 3.1 as the initial conditions, we conduct two sets of assimilation experiments $WW3_{WW3(J2)}{}^{CFSR}$, $WW3_{WW3(J2+J3)}{}^{CFSR}$, $WW3_{WW3(J2+J3+SA)}{}^{CFSR}$ and $SWAN_{SWAN(J2)}{}^{CFSR}$, $SWAN_{SWAN(J2+J3)}{}^{CFSR}$, $SWAN_{SWAN(J2+J3+SA)}{}^{CFSR}$. Through examining the assimilation quality with one satellite (Jason-2), two satellites (Jason-2+Jason-3) and three satellites (Jason-2+Jason-3+SARAL), we attempt to understand

the impact of improving the observing system on the WDA, considering the NCEP-CFSR wind forcing errors against the ECMWF ERA-Interim based on a perfect assimilation model. The RMSEs of all the above assimilation experiments are plotted in **Fig. 3** as the blue (assimilating Jason-2 only), green (assimilating Jason-2+Jason-3) and cyan (assimilating Jason-2+Jason-3+SARAL) lines.





From **Fig. 3**, we can see that in both models, the assimilation errors are reduced when more observational information is used. The corresponding RMSE reductions in these three experiments from the model control run are 24 %, 32 % and 38 % for WW3 and 26 %, 35 % and 38 % for SWAN, respectively. However, when more satellite observations are assimilated into the model, the magnitude of improvement becomes small (further reduced by 8 % in WW3 and 9 % in SWAN when Jason-3 is added as well as only a 6 % in WW3 and 3 % in SWAN further reduction when SARAL is further added). These results suggest that given wind forcing errors, increasing observational information can help to improve the model behavior, but the improvement is limited.

### 3.3 Adverse impact of model bias

As described in section 2.2, the WW3 and SWAN models discretize the wave action governing equation with different physical processes, parameterization schemes and differencing schemes. These differences result in each wave model having its own distinguished characteristics. To study the adverse impact of the model bias on the wave assimilation, the biased twin experiments described in section 2.4.1 are used in this section, where the "truth" model and the assimilation model are different between WW3 and SWAN. For example, the $WW3_{SWAN(J2)}^{ERAI}$ (resp. $SWAN_{WW3(J2)}^{ERAI}$) experiment uses WW3 (resp. SWAN) as the assimilation model to assimilate the Jason-2 track point "observations," but the observed values are produced by SWAN (resp. WW3), and all models are forced by the ERA-Interim wind. All related experiments for the biased model framework are described in detail in **Table 2**.

The RMSEs and correlation coefficients produced by the all biased model assimilation experiments are plotted in **Fig. 4.** The black line in each panel represents the result of the WW3 model control forced by the ERA-Interim wind ($WW3^{ERAI}$) against the "truth" simulation by the SWAN model with the same wind forcing ($SWAN^{ERAI}$) (left panels, a and b) (vice versa in the right panels, c and d). Both the $WW3^{ERAI}$ and $SWAN^{ERAI}$ experiments are initialized from a cold start by the wind and integrated for 80 days, and the results of the last 40 days are shown in **Fig. 4**. It is clear that the WW3 and SWAN model simulations are quite different even though both simulations use identical forcings and start from identical initial conditions. The RMSEs of the two model simulations are both 0.58 m, which is much larger than the errors produced by a perfect model but with different wind (~0.34 m, see **Fig. 3**).

Compared with the model controls $WW3^{ERAI}$ and $SWAN^{ERAI}$, the assimilation experiments $WW3_{SWAN(J2)}^{ERAI}$ and $SWAN_{WW3(J2)}^{ERAI}$ (pink lines in **Fig. 4**) can significantly reduce the SWH simulation





error (by 24 % and 22 %, respectively) and enhance the correlations (by 3 % and 4 %, respectively)
with the "truth" ($SWAN^{ERAI}$ and $WW3^{ERAI}$, respectively). When the "observations" of Jason-3 and
SARAL are added to the assimilation (i.e., $WW3_{SWAN(J2+J3)}^{ERAI}$ and $WW3_{SWAN(J2+J3+SA)}^{ERAI}$ as well as
$SWAN_{WW3(J2+J3)}^{ERAI}$ and $SWAN_{WW3(J2+J3+SA)}^{ERAI}$) (see the red and blue lines, respectively), the model
SWH error (resp. correlation) is further reduced (resp. enhanced), but the amplitude of reduction (resp.
enhancement) gradually diminishes (10 % and 5 % for further error reduction and 1 % and 0.8 % for
further correlation enhancement in the WW3 assimilation; 10 % and 7 % for further error reduction and
1.7 % and 0.7 % for further correlation enhancement in SWAN assimilation from the additions of
Jason-3 and SARAL, respectively).

The results of two other sets of assimilation experiments called $WW3_{SWAN(J2)}^{ASSW}$ and $SWAN_{WW3(J2)}^{ASSW}$
are also plotted by dotted green lines in **Fig. 4.** The superscript "*ASSW*" stands for the assimilation-
corrected wind, meaning that the wind forcing of the assimilation model is also "corrected" by the
"observed" SWH data, as described in Step 4 of section 2.3. We found that in the WW3 model, using
the "observed" SWH data to "correct" the wind can compensate for the model errors to some degree
and further reduce the assimilation errors, but the improvement is very limited. In the SWAN model,
such wind "correction" cannot compensate for the model error at all.

These assimilation results clearly show that even though the wind forcing is perfect, once a biased
assimilation model is used, the wave simulation has large errors. Although WDA can greatly reduce the
simulation error by assimilating the observational information into the model, due to the existence of
the model bias, the error remains at some significant level and cannot be eliminated entirely even by
increasing the observational constraints through an improvement in the observational system and
constraint of the wind forcing. Next, with the results of real-data assimilation where both the model and
wind forcing have errors, we will analyze and discuss how to mitigate the model bias influences on the
WDA.

## 4 Mitigation of model bias influences on wave assimilation

### 4.1 Bias characteristics of WW3 and SWAN data assimilations

From the above analyses of twin experiment results, we learned that the model bias has a strong
adverse impact on WDA. To explore the method of mitigating the model bias influence on the WDA,
we conduct the real-data assimilation experiments (same time range as the "twin" experiments)





described in section 2.4.2 using the WW3 and SWAN assimilation systems to assimilate the track data of Jason-2 and Jason-2+Jason-3+SARAL. To ensure the performance of the biased model WDA, a

longer assimilation (more than two months) is conducted (a total of 70 days). The spatial distributions of the SWH errors (verified against the merged gridded AVISO observations over the last 30 days out of 80 days) are shown in **Fig. 5**. The left (resp. right) panels are for the WW3 (resp. SWAN) simulation and assimilations: $WW3^{ERAI}$, $WW3_{J2}^{ERAI}$, and $WW3_{J2+J3+SA}^{ERAI}$ (resp. $SWAN^{ERAI}$, $SWAN_{J2}^{ERAI}$, and $SWAN_{J2+J3+SA}^{ERAI}$).

Comparing panel a with panel d in **Fig. 5** reveals that a large difference exists in the simulations of the two models. First, the SWAN simulation errors are generally larger than the WW3 simulation errors. Second, the global error distributions are quite different: while the WW3 simulation errors appear negative (resp. positive) over most of the 30° S north (resp. south) area, the SWAN simulation errors appear positive in most of the tropical oceans and negative in the middle latitudes. Both simulations

show large errors in the southern ocean coastal area, but over the Antarctic Circumpolar Current area, the WW3 error is positive, and the SWAN error is negative. Large model errors usually occur in strong wind areas. Although the same parametric scheme of wind input expressed by exponential growth with friction velocity can be activated, the process transforming wind speed provided by users to friction velocity is necessary but different in both models, which may be an important reason why the two

models have different performances under the same wind conditions.

The above systematic differences between the two model simulations have significant influences on the results of the WDA. In general, the distribution of assimilation errors shares the same patterns as the model simulation errors but with a much smaller magnitude. The net result is that both the WW3 negative (resp. positive) error magnitude over the 30ºS north (resp. south) area and the SWAN error

magnitude as (+)(-)(+)(-) from south to north are dramatically reduced by the Jason-2 data assimilation (comparing the middle panels with the upper panels), and on this basis, incorporating more observations from Jason-3 and SARAL into the assimilation process, both model error magnitudes are further reduced to some degree (comparing the bottom panels with the middle panels). From the corresponding RMSE distributions (**Fig. 6**), we learned that the large RMSEs mainly appear in places

where the model bias (time mean error) is large. This finding means that the model bias has a largely adverse impact on the WDA.

**Figure 7** displays the time series of the RMSEs and correlation coefficients with the statistics in space. The RMSE (resp. correlation coefficient) of the SWAN model simulation is larger (resp. smaller) than



the WW3 model simulation (.66 m RMSE and .806 correlation for SWAN versus .61 m RMSE
and .876 correlation for WW3). In the WW3 and SWAN assimilations with the Jason-2 data, the
RMSEs are reduced by 8 % and 11 %, respectively, and the time mean correlations are enhanced by
roughly 1 % and 5 %, respectively. If the data of all three satellites, Jason-2, Jason-3 and SARAL, are
assimilated, the RMSEs are reduced by 11 % and 17 % in the WW3 and SWAN assimilations,
respectively, and the correlations are enhanced by approximately 2 % and 8 %, respectively. In these
real-data assimilation cases, for each model assimilation, both the model and wind forcing have errors.
Under this circumstance, the assimilation where the SWH observations are used to adjust the model
spectrum, the model wind forcing is also corrected and can further reduce the assimilation errors (red
lines in **Fig. 7**). The red lines in **Fig. 7** represent the best result of the assimilation given the WW3 and
SWAN model biases, which makes full use of the observations from all three satellites to adjust both
the model spectrum and wind forcing. Next, we will discuss how to use the results of two assimilation
systems to mitigate the wave analysis error.

**4.2 Mitigation of WDA errors**

The mitigation of model bias is a complex issue in which improving the model is a final but long-
lasting solution. From **Fig. 5,** we learned that the WW3 and SWAN assimilation errors have some
common (or opposite) characteristics in some locations. For example, while the SWH over the southern
ocean coastal area always appears to be overestimated because of the lack of adequate observations to
improve in both assimilation systems, the WW3 and SWAN assimilations appear to be the opposite in
the Antarctic Circumpolar Current area and the tropical oceans. The WW3 (resp. SWAN) assimilation
errors in the Antarctic Circumpolar Current area appear positive (resp. negative), while the WW3 (resp.
SWAN) assimilation errors in the tropical oceans appear negative (resp. positive). A question arises: as
the first step of mitigating model bias influences on the WDA, can we use a pair of assimilation
systems to explore a statistical approach to reduce the wave assimilation errors?
Given the opposite behaviors of two assimilation systems existing in certain places, the simplest bias
correction can be conducted by simple average. This assumes that bias itself has a Gaussian
distribution with trivial expectation. The corresponding results are shown in **Fig. 8**. Compared with the
performance of each individual assimilation system (dashed red lines for WW3 and dotted red lines for
SWAN), the results of this bias correction (cyan lines) show that the RMSE is reduced (the left panel)
but the correlation is not greatly improved (the right panel). It is reasonable that based on the opposite





errors deviating from the real world in two assimilation systems, this correction method employing the
mathematical average can reduce the RMSE to some extent, but it may not have a significant
contribution to improving the correlation coefficient if either the sampling size of model bias is too
small (only 2 in this case) or the bias has an asymmetric distribution.

Considering the potentially asymmetric distribution of bias (i.e., each wave model has its own
characteristics of systematic error due to deficit physics) and small sampling size in practice, we
calculate the bias (i.e., long-term mean of the error) for each assimilation system and extract it first and
then calculate the expectation (average). The corresponding results are shown with pink lines in **Fig. 8**.
Both the RMSE and correlation are improved greatly. Clearly, this bias correction with physical
consideration is more effective to improve the quality of WDA, but it uses observational information
one more time, while the first method of bias correction processes assimilation results directly without
further uses of observational information.

To verify the feasibility and applicability of the bias correction method above, 3 well-known wave
models (WW3, SWAN and WAM) with the same data assimilation method are used to conduct longer
assimilation and bias correction experiments. The calculation period lasts for 14 months (from
November 2016 to December 2017) with sufficient spin-up process to reach a steady assimilation state
(the 1st month for model spin-up and the 2nd month for assimilation spin-up). The results of the last 12
months (for 2017) are analyzed and presented in **Fig. 9** and **Fig. 10** for the spatial distributions and
time series of RMSEs and correlation coefficients, respectively. From **Fig. 9**, we can see that both the
RMSE and correlation coefficient (panels d and h, respectively) have been improved by the bias
correction that combines the advantages of every WDA system (panels a and e for WW3, panels b and
f for SWAN, panels c and g for WAM). In **Fig. 10**, the bias correction of model control runs shows
improvement but is worse than the data assimilation before bias correction (compare green with pink).
Compared with the model control (blue), the assimilation results with bias correction (red) can reduce
the error by 25 % and significantly enhance the correlation coefficient (from 0.88 to 0.923). This result
confirms that this bias correction based on multiple assimilation systems can effectively enhance the
WDA quality.



## 5 Summary and discussion

Ocean waves cause the sea surface roughness to impact the boundary conditions of the atmosphere and the wind stress of the ocean surface. Wave processes, such as wave-breaking, wave-induced bottom stress and so on, have significant effects on ocean mixing. Thus, ocean waves are important physical
processes for understanding ocean mixing and air-sea interactions in coupled Earth systems. More accurately predicting ocean waves is of great societal significance. However, multiple error sources exist in wave simulations and predictions, including modeling errors, wind forcing errors and initial condition errors.

To sort out the source of the errors of wave data assimilation (WDA), a pair of independent WDA
systems is first developed using two wave models: Wave Watch III (WW3) and Simulating WAves Nearshore (SWAN). The perfect and biased model "twin" experiment frameworks are designed to clearly identify each error source and examine its influences on WDA. The results show that model bias is a significant error source that has a largely adverse impact. Then, two WDA systems are used to design bias correction approaches to mitigate the influences of model bias and improve the assimilation
quality. Finally, long-term WDA experiments added by the third WDA system with the WAM model (WAve Modeling) (WW3, SWAN and WAM) are conducted to validate the bias correction method. Three findings are established: 1) When the model is perfect, the correct wind forcing can overwrite the initial condition error within a 10-day time scale, but the WDA can shorten the time scale by half. 2) When the model is biased, despite a perfect wind forcing, the wave simulation has large errors and the
WDA can only reduce the error to a limited extent. 3) With the results from two assimilation systems, a statistical approach of bias correction significantly improves the quality of final wave analysis by combining the merits from individual assimilation systems.

Model bias is an obstacle to improving WDA and wave predictions. Using multiple assimilation systems to study the influences of model bias on WDA is an effective approach. As the first step,
however, we have used a simple assimilation scheme and simple bias correction method. In follow-up studies, we shall consider advanced assimilation schemes and more comprehensive correction methods to help improve modeling. For example, the "online" bias correction during the assimilation process (e.g., Dee, 2005) will be considered to improve the assimilation results within individual assimilation systems. In addition, improving the model is an important, inevitable and long-lasting task. In this
study, we find that three models show common bias characteristics in the Antarctic Circumpolar Current (ACC) area. This finding suggests that present wave modeling may have deficits in energy





spectrum expression for high wind speed areas. In the future, we will further examine the sensitivities of physical processes on high wind speed to mitigate such common modeling bias. All in all, a robust bias correction method with lower model bias and higher representation of wave physical characteristics may further improve wave analysis quality. Once a long time series of high-quality wave analyses is available, it is expected that we can improve our understanding of ocean mixing. The physical process of wave-induced mixing is linked with the structure of the ocean mixing layer (Qiao et al., 2010). This process can be expressed as a function of wave number, frequency and wave spectrum and so on, provided by wave analysis. With the framework of multiple WDA systems developed in this study, improved wave predictions can be effectively pursued. How can we further enhance the predictability of ocean waves? The first important step is to understand the physical process of ocean waves better based on a more accurate evolution of wave state from this framework. Answering these questions could be very important and interesting research topics for the future studies.

**Code and data availability**

Codes, data and scripts used to run the models and produce the figures in this work are available on the Zenodo site: doi:10.5281/zenodo.3445580 or by sending a written request to the corresponding author (Shaoqing Zhang, szhang@ouc.edu.cn).

**Author contribution**

Jiangyu Li coded the module of data assimilation and inserted it into the wave models, carried out all the experiments. Shaoqing Zhang designed the experiments, analysed the results with constructive discussions. Both authors wrote the paper together.

**Competing interests**

The authors declare that they have no conflict of interest.





## Acknowledgments

The research is supported by the National Key R&D Program of China (2017YFC1404100 and 2017YFC1404102) and the National Natural Science Foundation of China (Grant No. 41775100, 41830964).

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

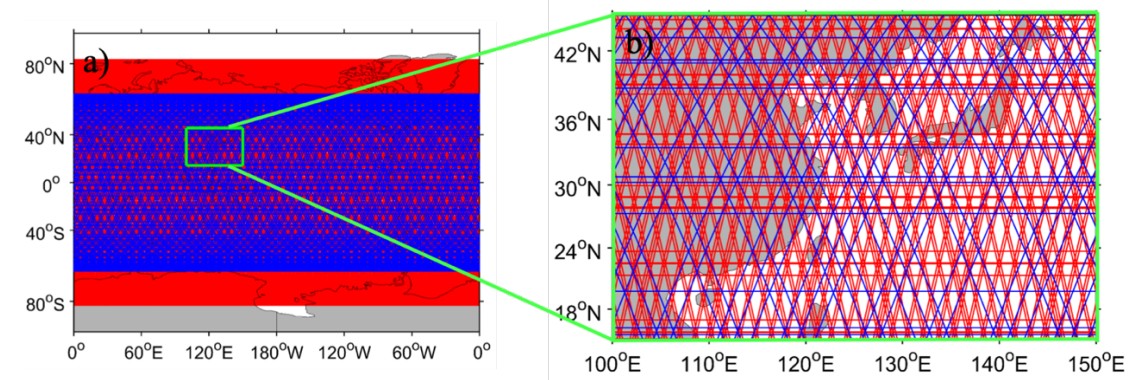

**Figure 1: The ground projection tracks of satellite Jason-2 (blue line, with an inclination of 66° N(S)) and SARAL (red line, with an inclination of 88° N(S)) in one cycle over approximately 10 days and 35 days, respectively, in the a) global and b) East Asia domains (zoomed out of green box in panel a).**


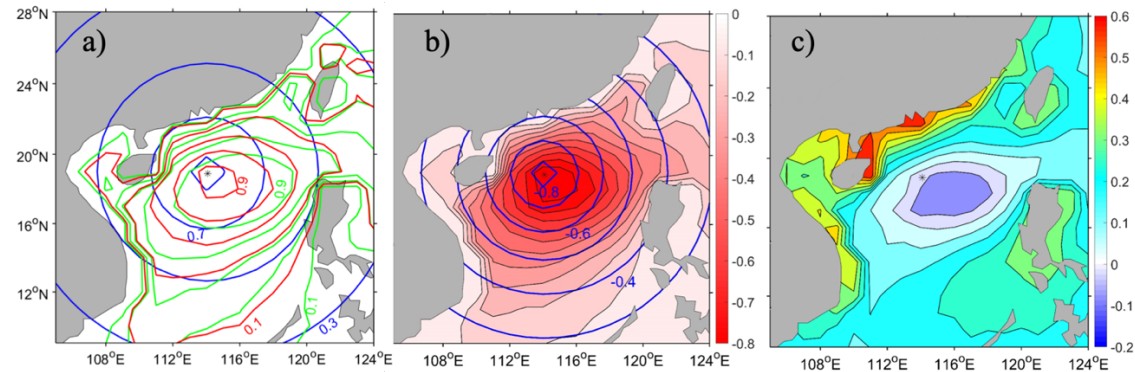

**Figure 2: Spatial distributions of a) background correlation coefficients by the empirical correlation model (blue), model data statistics (green) as well as their combination of Eq. (3) (red), b) adjustment increments of SWH, and c) the SWH difference from panel b by an analysis process given the single observation obtained at 114.09° E, 18.90° N, denoted by the asterisk (unit: meter).**



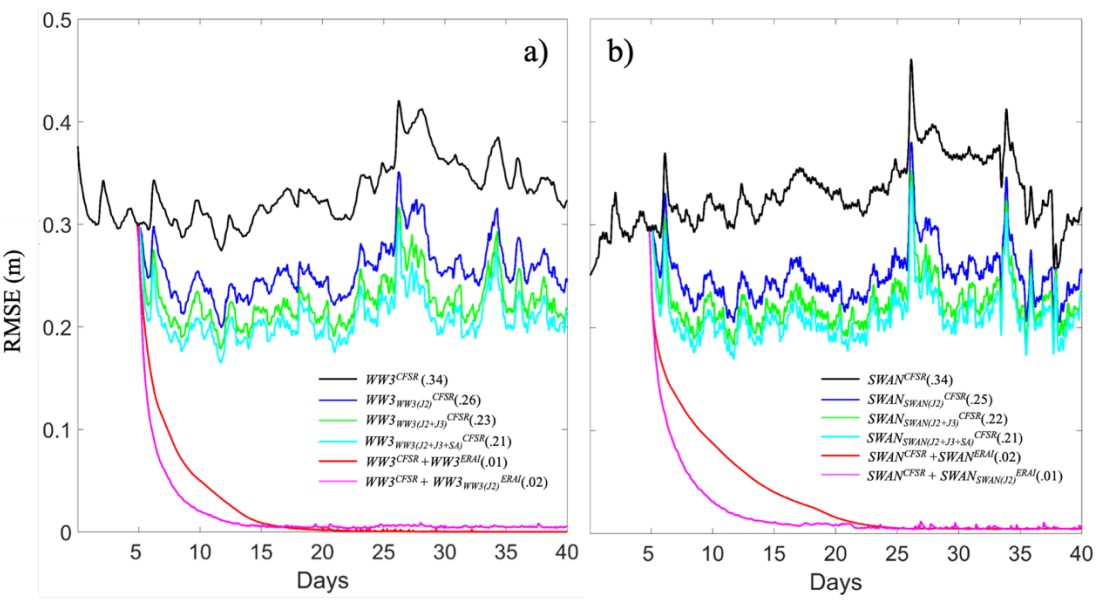


**Figure 3: The time series of RMSEs of the a) WW3 and b) SWAN perfect model experiments in the model control run with the NCEP CFSRv2 wind (black, denoted as WW3$^{CFSR}$ in panel a and SWAN$^{CFSR}$ in panel b), assimilating the "observed" data sampled by the tracks of Jason-2 (blue, denoted as WW3$_{WW3(J2)}^{CFSR}$ and SWAN$_{SWAN(J2)}^{CFSR}$), Jason-2 & 3 (green, denoted as WW3$_{WW3(J2+J3)}^{CFSR}$ and SWAN$_{SWAN(J2+J3)}^{CFSR}$), as**

**well as Jason-2 & 3 and SARAL (cyan, denoted as WW3$_{WW3(J2+J3+SA)}^{CFSR}$ and SWAN$_{SWAN(J2+J3+SA)}^{CFSR}$) against the "truth" simulation forced by the ERA-Interim wind. The red and pink are forced by the NCEP CFSRv2 wind in the first 45 days, but the next 35 days are forced using the ERA-Interim wind (same as "truth") without (denoted as WW3$^{CFSR}$ +WW3$^{ERAI}$ and SWAN$^{CFSR}$ +SWAN$^{ERAI}$) or with (denoted as WW3$^{CFSR}$ + WW3$_{WW3(J2)}^{ERAI}$ and SWAN$^{CFSR}$ + SWAN$_{SWAN(J2)}^{ERAI}$) the assimilation of Jason-2 data. The**

**number in parentheses for each color is the corresponding RMSE summed over the verification time period (30 days after the 45-day model spin-up and 5-day assimilation spin-up). The "observed" data are produced by projecting the "truth" SWH onto the satellite orbit.**



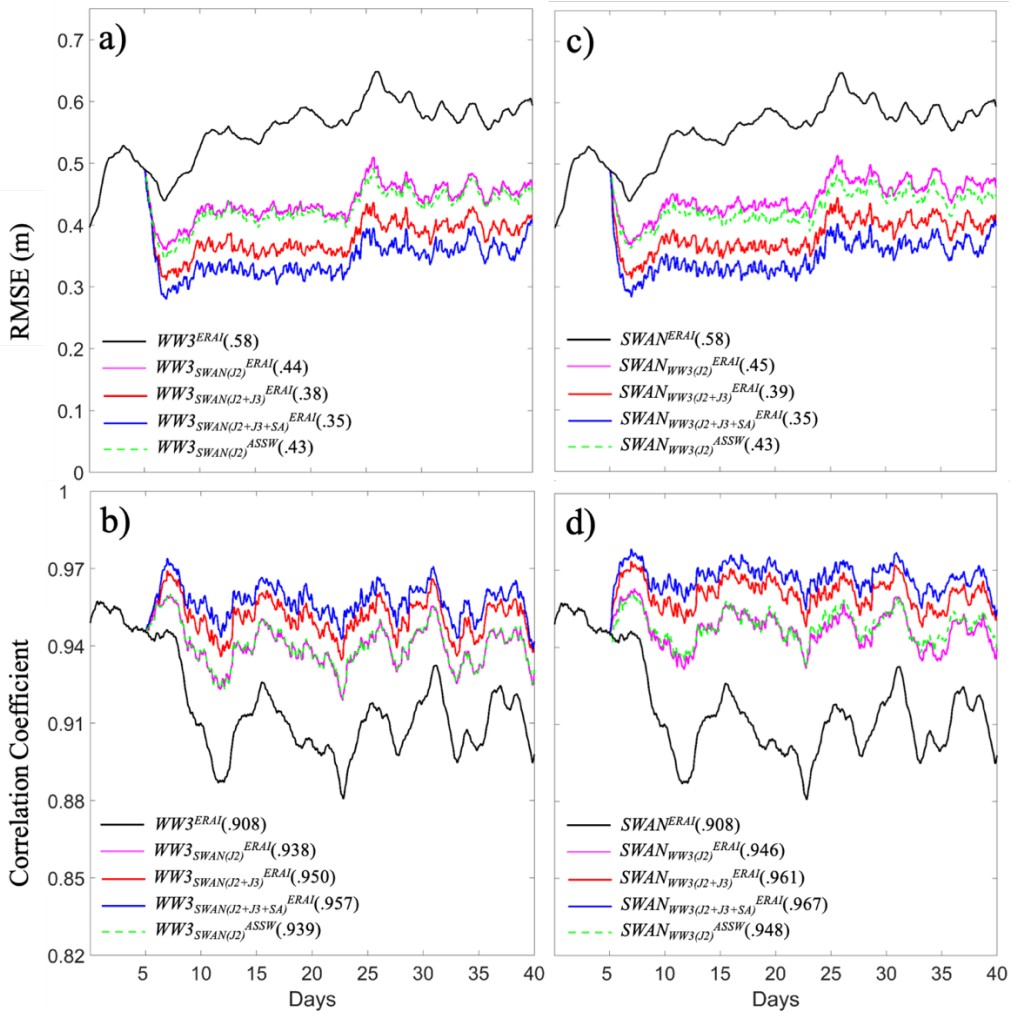

**Figure 4: The time series of RMSEs (upper) and correlation coefficients (bottom) of the WW3(left) and SWAN(right) biased model experiments in the model control run forced by the ERA-Interim wind (black, denoted as WW3$^{ERAI}$ and SWAN$^{ERAI}$), assimilations with "observed" data from one (pink, denoted as WW3$_{SWAN(J2)}^{ERAI}$ and SWAN$_{WW3(J2)}^{ERAI}$), two (red, denoted as WW3$_{SWAN(J2+J3)}^{ERAI}$ and SWAN$_{WW3(J2+J3)}^{ERAI}$), and three (blue, denoted as WW3$_{SWAN(J2+J3+SA)}^{ERAI}$ and SWAN$_{WW3(J2+J3+SA)}^{ERAI}$) satellites, as well as the assimilation with corrected wind (dotted green, denoted as WW3$_{SWAN(J2)}^{ASSW}$ and SWAN$_{WW3(J2)}^{ASSW}$) against the "truth" (same as in Fig. 3 but for SWAN and WW3 with the ERA-Interim wind). The numbers in the parentheses correspond to the RMSE (in panels a and c) and correlation coefficient (in panels b and d) over the last 30 days during the assimilation period. The "observed" data are produced by projecting the "truth" SWH onto the satellite orbit.**



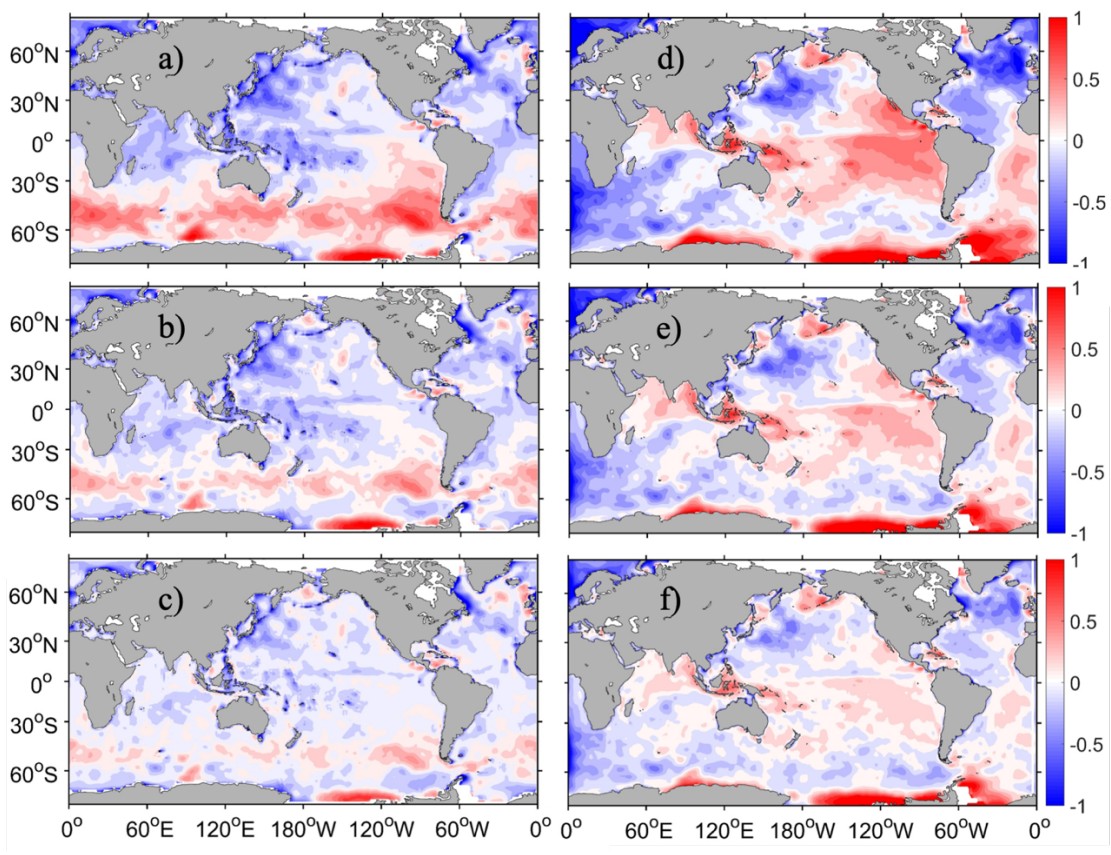

**Figure 5: Distributions of SWH mean errors (against the merged grid altimeter data) of the WW3 (left) and SWAN (right) model simulations (upper) and assimilations with Jason-2 (middle), as well as all Jason-2, Jason-3 and SARAL (bottom) data forced by the ERA-Interim wind. The statistics are conducted over the last 30 days of a 70-day total assimilation period (unit: meter).**

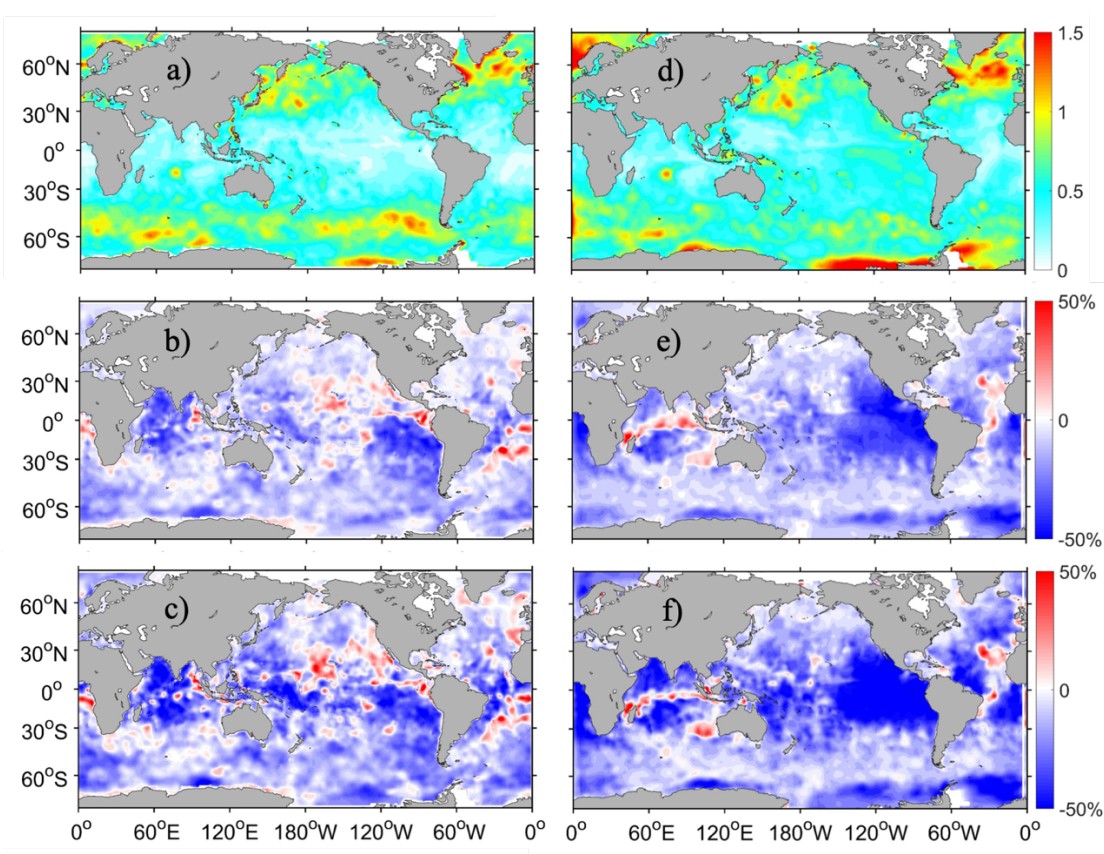

**Figure 6: Same as Fig. 5 but for the RMSEs. Panels b, c, e and f show the percentage of RMSE reduction from the model control (unit: meter).**




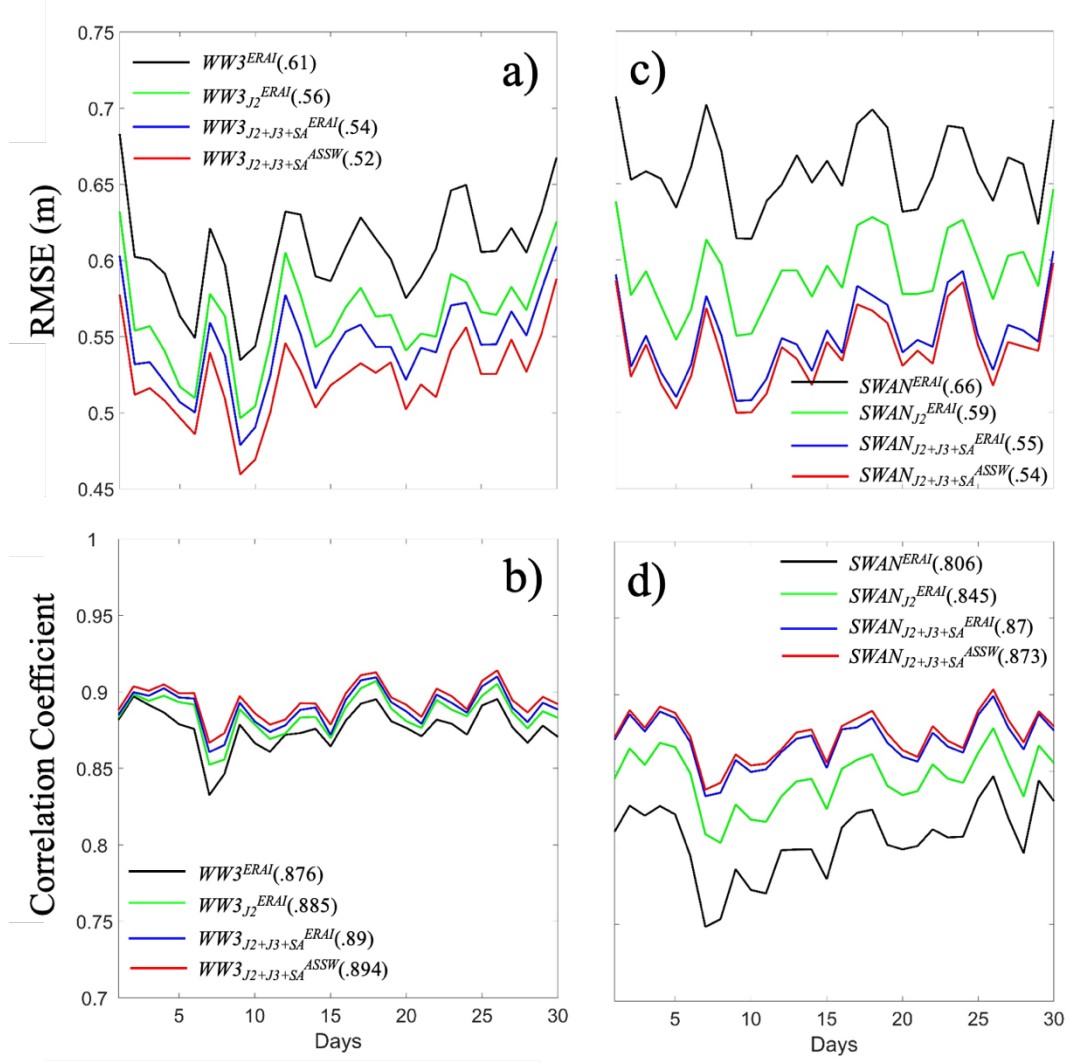

**Figure 7: Time series of RMSEs (upper) and correlation coefficients (bottom) of WW3 (left) and SWAN (right) produced by the model control run (black, denoted as WW3$^{ERAI}$ and SWAN$^{ERAI}$), assimilation using the data from one (green, denoted as WW3$_{J2}^{ERAI}$ and SWAN$_{J2}^{ERAI}$) and three satellites (blue,**
**denoted as WW3$_{J2+J3+SA}^{ERAI}$ and SWAN$_{J2+J3+SA}^{ERAI}$) with corrected wind (red, denoted as WW3$_{J2+J3+SA}^{ASSW}$ and SWAN$_{J2+J3+SA}^{ASSW}$).**



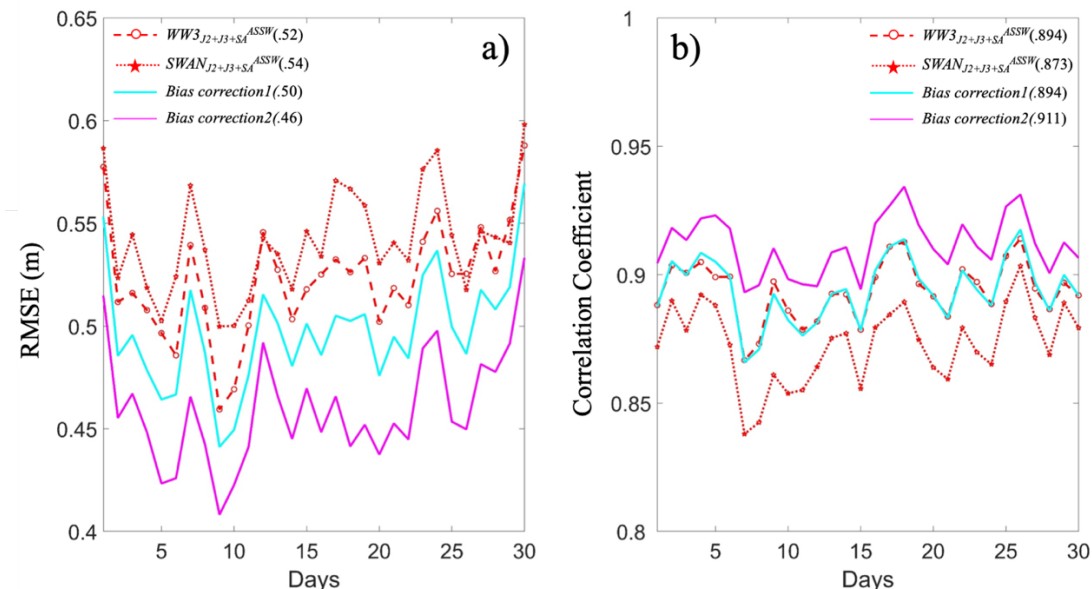

**Figure 8: Time series of a) RMSEs and b) correlation coefficients produced by two bias correction schemes**
**(cyan and pink) through a combination of WW3 and SWAN assimilations with the data from three**
**satellites (Jason-2, Jason-3 and SARAL) and wind correction starting from the ERA-Interim wind. The**
**results of the individual assimilation systems are plotted as dotted and dashed red lines (taken from Fig. 7)**
**for reference.**



**Figure 9: Distributions of SWH RMSEs (left) and correlation coefficients (right) (against the merged grid altimeter data) of the WW3 (a, e), SWAN (b, f) and WAM (c, g) assimilations and the bias correction (d, h). The statistics are conducted over the entire 1-year assimilation period. Other information is the same as that in Fig. 8.**



**Figure 10: The time series of a) RMSEs and b) correlation coefficients produced by the model control run (blue), data assimilation (pink) and their corresponding bias corrections (green and red) combining three wave model assimilation results (WAM, WW3 and SWAN) over 1 year. Other information is the same as in Fig. 8.**





**Table 1: List of perfect model twin experiments.**

| Exp. name | Model | Wind force | Assimilation or not | Role |
|---|---|---|---|---|
| $WW3^{ERAI}$ | WW3 | ERA-Interim | No | Truth for WW3 assimilation |
| $WW3^{CFSR}$ | WW3 | NCEP-CFSR | No | Model control for WW3 assimilation reference |
| $WW3_{WW3(J2)}^{CFSR}$ | WW3 | NCEP-CFSR | Yes (using Jason-2 track) | Impact of observational system |
| $WW3_{WW3(J2+J3)}^{CFSR}$ | WW3 | NCEP-CFSR | Yes (using tracks of Jason-2 and Jason-3) | |
| $WW3_{WW3(J2+J3+SA)}^{CFSR}$ | WW3 | NCEP-CFSR | Yes (using tracks of Jason-2, Jason-3 and SARAL) | |
| $SWAN^{ERAI}$ | SWAN | ERA-Interim | No | Truth for SWAN assimilation |
| $SWAN^{CFSR}$ | SWAN | NCEP-CFSR | No | Model control for SWAN assimilation reference |
| $SWAN_{SWAN(J2)}^{CFSR}$ | SWAN | NCEP-CFSR | Yes (using Jason-2 track) | Impact of observational system |
| $SWAN_{SWAN(J2+J3)}^{CFSR}$ | SWAN | NCEP-CFSR | Yes (using tracks of Jason-2 and Jason-3) | |
| $SWAN_{SWAN(J2+J3+SA)}^{CFSR}$ | SWAN | NCEP-CFSR | Yes (using tracks of Jason-2, Jason-3 and SARAL) | |


**Table 2: List of biased model twin experiments.**

| Exp. name | Model | Wind source | Assimilation or not | Role |
|---|---|---|---|---|
| $WW3^{ERAI}$ | WW3 | ERA-Interim | No | Truth for SWAN assimilation |
| $SWAN^{ERAI}$ | SWAN | ERA-Interim | No | Model control for SWAN assimilation |
| $SWAN_{WW3(J2)}^{ERAI}$ | SWAN | ERA-Interim | Yes (using Jason-2 track) | Impact of observational system |
| $SWAN_{WW3(J2+J3)}^{ERAI}$ | SWAN | ERA-Interim | Yes (using tracks of Jason-2 and Jason-3) | |
| $SWAN_{WW3(J2+J3+SA)}^{ERAI}$ | SWAN | ERA-Interim | Yes (using tracks of Jason-2, Jason-3 and SARAL) | |
| $SWAN_{WW3(J2)}^{ASSW}$ | SWAN | Assimilation-corrected wind based on ERAI | Yes (using tracks of Jason-2) | Impact of assimilation-corrected wind |
| $SWAN^{ERAI}$ | SWAN | ERA-Interim | No | Truth for WW3 assimilation |
| $WW3^{ERAI}$ | WW3 | ERA-Interim | No | Model control for WW3 assimilation reference |
| $WW3_{SWAN(J2)}^{ERAI}$ | WW3 | ERA-Interim | Yes (using Jason-2 track) | Impact of observational system |
| $WW3_{SWAN(J2+J3)}^{ERAI}$ | WW3 | ERA-Interim | Yes (using tracks of Jason-2 and Jason-3) | |
| $WW3_{SWAN(J2+J3+SA)}^{ERAI}$ | WW3 | ERA-Interim | Yes (using tracks of Jason-2, Jason-3 and SARAL) | |





| $WW3_{SWAN(J2)}{}^{ASSW}$ | WW3 | Assimilation-corrected wind based on ERAI | Yes (using tracks of Jason-2) | Impact of assimilation-corrected wind |
|---|---|---|---|---|
