# Peer review of "Mitigation of Model Bias Influences on Wave Data Assimilation with Multiple Assimilation Systems Using WaveWatch III v5.16 and SWAN v41.20"

_Geoscientific Model Development, 2019_

## Referee Comment (RC1) · Anonymous Referee #1 · 27 Nov 2019

This study investigate the challenge of model bias on wave model data assimilation. It apply a set of twin-model experiments to quantify the different error sources in wave model: initial, boundary, and model error. Based that It provide a simple statistical approach to reduce the impact of model bias and improve the assimilation results. The topic is interesting and important for wave data assimilation, well fit for GMD. The experiment are well designed and manuscript is in good shape. Here I only have few points to further polish this work. Therefore I suggest miner revision for current version.

1 There are three error sources: initial/boundary/model-bias. You have identify them in

your biased/unbiased twin experiment. It turns out both boundary and model-bias are important.

In my pointer of view, both error sources could lead to SWH biases/uncertainties for assimilation/simulation. It would be great to separate them and quantify the improvement percentage by your data assimilation from your biased/unbiased twin experiments. The SWH bias and its decrease percentage may give your hind on mitigation the assimilation bias for the real observation. Similar separation should apply to real observation assimilation cases. In your figure 5, mainly represent the SWH bias. You can recalculate figure 6 (RMSE) after remove the bias in figure 5, which represent the uncertainties related to boundary/model bias.

When you apply average to reduce the assimilation error of SWH. You only reduces the uncertainty part but not the bias part. You have to direct remove the bias from the reanalysis.

2 In your real observation assimilation, the boundary and model-bias are both included. You may compare spatial pattern and the decreasing percentage of SWH bias with those your biased/unbiased twin experiments to speculate which source (boundary/model-bias) has stronger impact in certain area..

3 You only applied one kind of wind forcing for different models and then use them to do the bias correction. Since the bias also come from the boundary forcing. I encourage you using two kinds of wind forcing to further investigate the bias/uncertainty generated by the boundary forcing.

4 it is unclear how do you get the SWH bias for bias correction, what is the spatial pattern. I thought it should refer to the figure 5, but I did not find those in the manuscript.

5 Model description in page 4. This part need be more condensed with appropriate reference. Readers would appreciate more on the differences among those models, instead of the comparison to their own previous version. You may highlight the advantage or disadvantage among three modes.

---

## Referee Comment (RC2) · Anonymous Referee #2 · 9 Dec 2019

This paper describes the effect of OI data assimilation on ocean wave simulations, including bias correction. The work is mainly done through running a series of twin experiments. The paper contains scientifically interesting results, clear writing and good quality of figures. There is a potential to use the system to produce a global wave reanalysis (ensemble) product. There are some questions and comments, which are descried as following.

In Model configurations: the resolution of wind forcing could cause misleading. For example, 0.125x0.125deg is not the resolution of ERA-Interim itself (in which the model

resolution is much coarser), but is the resolution of gridded reanalysis data. Please clarify.

AVISO data: along-track data are used in wave simulation and gridded data are used for validation. What will be the possible effect due to difference of these data? Are there any observation-related errors/uncertainties that would influence the validation results? Also, the validation data are not independent to those assimilated. Do author consider using wave buoy data for validation in the future?

Following above comment, now some latest observation data contain wave direction information (i.e. peak wave direction, 2D spectra). These obs can be used to assimilate model wave spectra, which will have more advantages than assimilating SWH only. This might be worth mentioning somewhere in the paper.

L190: why did authors choose Sigma_M=0.6m and sigma_O=0.25m? Are the same model and observations as used in (Qi and Fan, 2013)? Is sigma varying with time, space and models? There ought to be some assumptions before using these parameters.

In equation 3, sigma_i/sigma_k*r_i,k is for SWH (or wave spectra?) correlation and statistics. I just wonder whether wave covariances will have the same structure as wave error covariances as equation 3 is supposed to be for error covariance. In a storm the high-sea state may have a few hundred km long, but this doesn't necessarily mean the error is propagating in a few hundred km distance. I don't have a solution for this question. But it needs some assumptions on equation 3, with clarifications of potential drawbacks, before using it.

2.3 section step 4: ocean waves have two components, i.e. windsea waves and swells. The wind should (only) be corrected based on the analysed windsea waves, while analysed swells that are not directly forced by local wind have no impact on wind correction. This concept is described in Lionello et al 1995. Mostly wave models can output windsea and swell SWH. Why did not authors use the windsea SWH (rather than use total

SWH) to correct wind forcing? Using analysed total SWH to correct wind, wind could be overly corrected for example when it is a swell dominant event at the DA time.

Fig2: not very clear what b and c are for in these snapshots. SWH difference between what? Which is for increment? Please clarify.

Fig3,4: are these statistics for global mean or any regions? In Fig4, what are the correlation coefficients for (spatial correlations)? Same for other figures.

Fig4 shows wind correction does not clearly improve SWH simulation when assimilating J2. What about assimilating all satellite tracks i.e. J2+J3+SA? Does wind correction have a stronger effect?

Section 4.1 and Fig7: results show that wind correction only improves wave simulation by certain degrees. 1) Does the wind correction scheme used here have an impact? Can authors show (or suggest) any difference when using the scheme of Lionello et al 1995 (see above comment); 2) How about the spatial distribution of Fig7 red lines? 3) Are there more improvements seen in windsea waves than in swells? (you can simply partition wind sea waves and swell waves from total SWH). I assume wind correction will have a stronger impact in windsea wave simulations at high latitudes with strong wind.

Section 4.2: Please describe a bit more how the bias is produced and removed in these simulations. It is not very clear to me. How was 'bias correction of model control run' implemented?

Was bias correction in this paper like the offline-type bias correction? If we can have a long-term historic run, to produce the climatology of wave bias, and then use it as an offline bias correction term before online DA term (simply like some DA procedures in European systems), will this produce similar results as produced in Fig 10? This offline term can potentially be used in forecast as well. One normally won't expect that DA can efficiently correct the long-term and persistent bias, but expects DA is

more powerful for correcting the instantons/short-term/flow-dependent errors. It is not simple to have an immediate answer for this question, but it will be useful to have a discussion somewhere in the paper.

Line 61: to produce=> for producing L337: inaccurate

———————————————

---

## Author Comment (AC1) · 7 Jan 2020

In general:

The comments and suggestions from Reviewer #1 are excellent, which are very helpful for us to compare the influence of error sources more clearly. In the revision, we have added: (1) section 3.4 to quantify the comparison of the wind forcing error and model error in time and space, (2) clearer description about how to get SWH (significant wave height) bias and bias correction.

The following is the point-by-point reply to address the comments and suggestions of Reviewer #1.

This study investigates the challenge of model bias on wave model data assimilation. It applies a set of twin-model experiments to quantify the different error sources in wave model: initial, boundary, and model error. Based that It provide a simple statistical approach to reduce the impact of model bias and improve the assimilation results. The topic is interesting and important for wave data assimilation, well fit for GMD. The experiments are well designed and manuscript is in good shape. Here I only have few points to further polish this work. Therefore, I suggest miner revision for current version.

RE: Thanks for reviewer's encouragement.

1 There are three error sources: initial/boundary/model-bias. You have identify them in your biased/unbiased twin experiment. It turns out both boundary and model-bias are important. In my pointer of view, both error sources could lead to SWH biases/uncertainties for assimilation/simulation. It would be great to separate them and quantify the improvement percentage by your data assimilation from your bi-ased/unbiased twin experiments. The SWH bias and its decrease percentage may give your hind on mitigation the assimilation bias for the real observation. Similar sep-aration should apply to real observation assimilation cases. In your figure 5, mainly represent the SWH bias. You can recalculate figure 6 (RMSE) after remove the bias in figure 5, which represent the uncertainties related to boundary/model bias. When you apply average to reduce the assimilation error of SWH. You only reduces the uncer-tainty part but not the bias part. You have to direct remove the bias from the reanalysis.

RE: Thanks for your thoughtful advice. We have added the quantitative comparison between boundary and model bias, please see lines 443-449 for their performances in the twin experiments. In the experiments of real-data assimilation, we also have re-plotted the Fig. 6 (Fig. 9 in the revised version) after removing the model bias shown

in Fig. 5 (Fig. 6 in the revision). The reviewer is right, as a try, the first method of bias correction is used to reduce the uncertainty part from the reanalysis and the second method is used to reduce the bias part.

2 In your real observation assimilation, the boundary and model-bias are both included. You may compare spatial pattern and the decreasing percentage of SWH bias with those your biased/unbiased twin experiments to speculate which source (boundary/model-bias) has stronger impact in certain area..

RE: Thanks for your excellent suggestion. In the twin experiments, we have compared the spatial pattern of errors caused by wind forcing and model bias, please see lines 458-476 and Fig. 5.

3 You only applied one kind of wind forcing for different models and then use them to do the bias correction. Since the bias also come from the boundary forcing. I encourage you using two kinds of wind forcing to further investigate the bias/uncertainty generated by the boundary forcing.

RE: Thank you for your helpful comment. In the real experiment, it's very difficult to distinguish the error sources. Then, as an understanding, we have tested the influences of these two error sources in the twin experiments. Next, we use a better wind forcing from ERA-Interim reanalysis in this manuscript to drive wave models and conduct bias correction in the real experiments. It is more reasonable and indicative to use bias correction mitigating hybrid errors and finally improve the quality of data assimilation results in the case of assimilating real observations.

4 it is unclear how do you get the SWH bias for bias correction, what is the spatial pattern. I thought it should refer to the figure 5, but I did not find those in the manuscript.

RE: Thanks for your kind reminder. More detailed description about how to get SWH bias has added in the revision, please see lines 556-561 and Fig. 6.

5 Model description in page 4. This part need be more condensed with appropriate
reference. Readers would appreciate more on the differences among those models, instead of the comparison to their own previous version. You may highlight the advantage or disadvantage among three modes.

RE: Thanks for your good advice. We simplified the description of this section, please see lines 112-119.

[Figure]

[Figure]

**Fig. 1.** figure5

**Fig. 2.** figure6

**Fig. 3.** figure9

---

## Author Comment (AC2) · 7 Jan 2020

In general:

The comments and suggestions from Reviewer 2 are very constructive, which are helpful for us to improve this manuscript. In this revision, we have added: 1) clearer captions of figures, 2) more detailed description for the assumption of equation in section 2.3, 3) more detailed description about how to get SWH bias and bias correction, 4) more discussions on "online-type" bias correction, the improvement of windsea and swell after

wind correction etc.

The following is the point-by-point reply to address the comments and suggestions of Reviewer 2.

This paper describes the effect of OI data assimilation on ocean wave simulations, including bias correction. The work is mainly done through running a series of twin experiments. The paper contains scientifically interesting results, clear writing and good quality of figures. There is a potential to use the system to produce a global wave reanalysis (ensemble) product. There are some questions and comments, which are descried as following.

RE: Thanks for your constructive comments, which have been fully addressed in the revision.

In Model configurations: the resolution of wind forcing could cause misleading. For example, 0.125x0.125deg is not the resolution of ERA-Interim itself (in which the model resolution is much coarser), but is the resolution of gridded reanalysis data. Please clarify.

RE: Thanks for your reminder. Clarified. Please see line 131.

AVISO data: along-track data are used in wave simulation and gridded data are used for validation. What will be the possible effect due to difference of these data? Are there any observation-related errors/uncertainties that would influence the validation results? Also, the validation data are not independent to those assimilated. Do author consider using wave buoy data for validation in the future?

RE: Thanks for your great comment. Along-track data is more effective to sample local wave information, while gridded dataset is an integration of multiple satellites, which more focus on the averaging variation over several days. Therefore, it is reasonable absorbing the along-track data into wave simulation in a small assimilation window with the assumption of little wave change in a short time. Nevertheless, it should be mentioned that due to a few satellites available, observations may lack representativeness in global area. At the same time, there is a potential validation error considering the obtained way of gridded data. As the reviewer said, the validation data are not fully independent to those assimilated. In the future, we will add buoy data to the verification, which is more reasonable and powerful to further illustrate the effect of bias correction. We have added discussions in the section 5. Please see lines 657-658.

Following above comment, now some latest observation data contain wave direction information (i.e. peak wave direction, 2D spectra). These obs can be used to assimilate model wave spectra, which will have more advantages than assimilating SWH only. This might be worth mentioning somewhere in the paper.

RE: Thanks for the good suggestion. We have added relevant description about observation variables and more discussions. Please see lines 481-484 and 652-654.

L190: why did authors choose Sigma_M=0.6m and sigma_O=0.25m? Are the same model and observations as used in (Qi and Fan, 2013)? Is sigma varying with time, space and models? There ought to be some assumptions before using these parameters.

RE: Thanks for your good advice. We have added clearer description of sigma. Please see lines 212-216.

In equation 3, sigma_i/sigma_k*r_i,k is for SWH (or wave spectra?) correlation and statistics. I just wonder whether wave covariances will have the same structure as wave error covariances as equation 3 is supposed to be for error covariance. In a storm the high-sea state may have a few hundred km long, but this doesn't necessarily mean the error is propagating in a few hundred km distance. I don't have a solution for this question. But it needs some assumptions on equation 3, with clarifications of potential drawbacks, before using it.

RE: Thanks for your thoughtful question. We have added the description of relevant

variables and the assumption of equation 3. Please see lines 224-227.

2.3 section step 4: ocean waves have two components, i.e. windsea waves and swells. The wind should (only) be corrected based on the analysed windsea waves, while analysed swells that are not directly forced by local wind have no impact on wind correction. This concept is described in Lionello et al 1995. Mostly wave models can output windsea and swell SWH. Why did not authors use the windsea SWH (rather than use total SWH) to correct wind forcing? Using analysed total SWH to correct wind, wind could be overly corrected for example when it is a swell dominant event at the DA time.

RE: This is an excellent suggestion. We have tested all corresponding experiments about correcting wind forcing with windsea wave height. However, the results do not show substantial improvement. The possible reason could be attributed to the simple correction method. It seems that in this simple correction method, it is difficult to distinguish the signal by using windsea wave height to correct wind from using the total SWH. In the future, we plan refine the correction method and study this part to enhance the signal-to-noise ratio. Please see lines 433-435 for added discussions on this point.

Fig2: not very clear what b and c are for in these snapshots. SWH difference between what? Which is for increment? Please clarify.

RE: Thank you for your reminder. We have added the detailed caption of Fig. 2. Please see lines 860-865.

Fig3,4: are these statistics for global mean or any regions? In Fig4, what are the correlation coefficients for (spatial correlations)? Same for other figures.

RE: Thanks for your good comments. We have added all relevant captions.

Fig4 shows wind correction does not clearly improve SWH simulation when assimilating J2. What about assimilating all satellite tracks i.e. J2+J3+SA? Does wind correction have a stronger effect?

RE: Thanks for your great suggestion. We have redisplayed the results of wind correction assimilating with three satellite tracks. Please see Fig. 4.

Section 4.1 and Fig7: results show that wind correction only improves wave simulation by certain degrees. 1) Does the wind correction scheme used here have an impact? Can authors show (or suggest) any difference when using the scheme of Lionello et al 1995 (see above comment); 2) How about the spatial distribution of Fig7 red lines? 3) Are there more improvements seen in windsea waves than in swells? (you can simply partition wind sea waves and swell waves from total SWH). I assume wind correction will have a stronger impact in windsea wave simulations at high latitudes with strong wind.

RE: Thanks for your thoughtful question. 1) In the real experiments, wind correction has a positive effect improving the assimilation results. However, different wave models have different improvement magnitudes. Here, SWAN has a weaker performance than WW3. Compared with the scheme in Lionello et al (1995), a big difference is that a coarse adjustment without distinguishing the wave structure (windsea or swell) is conducted to reconstruct the 2-dimensional wave spectrum. We will take this distinction into consideration in the future study. 2) We have displayed the spatial distribution of the red lines in Fig. 7, please see panel d and h of Fig. 9. 3) About the improvement of windsea and swell, we also have added discussions in the revision. Please see lines 582-588.

Section 4.2: Please describe a bit more how the bias is produced and removed in these simulations. It is not very clear to me. How was 'bias correction of model control run' implemented?

RE: Thanks for your great advice. We have added more detailed description about how to get bias and bias correction (please see lines 570-574). And we also have explained the "bias correction of model control run" (please see lines 615-617).

Was bias correction in this paper like the offline-type bias correction? If we can have a long-term historic run, to produce the climatology of wave bias, and then use it as

an offline bias correction term before online DA term (simply like some DA procedures in European systems), will this produce similar results as produced in Fig 10? This offline term can potentially be used in forecast as well. One normally won't expect that DA can efficiently correct the long-term and persistent bias, but expects DA is more powerful for correcting the instantons/short-term/flow-dependent errors. It is not simple to have an immediate answer for this question, but it will be useful to have a discussion somewhere in the paper.

RE: Thanks for your excellent suggestion. As the reviewer said, bias correction in this manuscript is an "offline-type". We have 1-year output of model control run and with data assimilation, then do offline-type bias correction and show in Fig. 11. If a longer run is conducted, similar results are supposed to display like Fig. 11. We will verify it in the future study. The effect of data assimilation on correcting the instantons errors is discussed in lines 654-658.

Line 61: to produce=> for producing L337: inaccurate

RE: Thanks for your advice. We have modified this error, please line 375.
* * *
[Figure]

[Figure]

**Fig. 1.** figure4

**Fig. 2.** figure9